# Graphene Transfer: A Physical Perspective

**DOI:** 10.3390/nano11112837

**Published:** 2021-10-25

**Authors:** Xavier Langston, Keith E. Whitener

**Affiliations:** Chemistry Division, US Naval Research Laboratory, 4555 Overlook Ave. SW, Washington, DC 20375, USA; xlangsto@xula.edu

**Keywords:** graphene, chemical vapor deposition, transfer, adhesion, electronics, fabrication

## Abstract

Graphene, synthesized either epitaxially on silicon carbide or via chemical vapor deposition (CVD) on a transition metal, is gathering an increasing amount of interest from industrial and commercial ventures due to its remarkable electronic, mechanical, and thermal properties, as well as the ease with which it can be incorporated into devices. To exploit these superlative properties, it is generally necessary to transfer graphene from its conductive growth substrate to a more appropriate target substrate. In this review, we analyze the literature describing graphene transfer methods developed over the last decade. We present a simple physical model of the adhesion of graphene to its substrate, and we use this model to organize the various graphene transfer techniques by how they tackle the problem of modulating the adhesion energy between graphene and its substrate. We consider the challenges inherent in both delamination of graphene from its original substrate as well as relamination of graphene onto its target substrate, and we show how our simple model can rationalize various transfer strategies to mitigate these challenges and overcome the introduction of impurities and defects into the graphene. Our analysis of graphene transfer strategies concludes with a suggestion of possible future directions for the field.

## 1. Introduction

Graphene, a two-dimensional hexagonal carbon allotrope, boasts a number of electronic, mechanical, thermal, and optical properties that make it very attractive for incorporation into next-generation electronic and optical devices [1,2,3,4,5]. Graphene fabrication proceeds via a variety of processes, but only a few of these processes are suitable for commercial production of graphene-based electronics. In particular, epitaxial growth of graphene on silicon carbide and chemical vapor deposition (CVD) of single- or few-layer graphene on copper or nickel are two of the most promising techniques for high-yield synthesis of electronics-quality graphene [6,7].

Although both epitaxial and CVD growth can produce high-quality single-layer graphene, it is often desirable or even necessary to transfer graphene from its original growth substrate onto a different substrate for the material to be useful in device applications. This is most apparent with CVD-grown graphene, where the graphene sits atop a metal substrate. A conductive, opaque metal substrate precludes many applications and therefore must be exchanged for another substrate which is more practical (usually an insulator or semiconductor for electronics applications and a transparent material for optical applications). Epitaxial growth does not have this problem explicitly, as SiC is an insulator. However, due to uneven growth patterns and step edges on epitaxially grown graphene, it is also often desirable to consider transfer of graphene to an more suitable substrate [8].

In this review, we present current knowledge concerning the transferring of graphene from one substrate to another. A number of reviews have appeared recently which focus on various aspects of the graphene transfer process [9,10], including maintaining the cleanliness of graphene [11,12], the nature of a polymer support layer [13], and prospects for industrial scale-up [14]. The present review focuses on graphene transfer with an eye toward the specific physical mechanisms and phenomena that influence the quality of the transfer. This review therefore does not seek to exhaustively cover every report on graphene transfer, but rather to illuminate important conceptual advances in the science of the graphene/substrate interaction, with particular focus on graphene transfer and device fabrication. Examining graphene transfer from a mechanistic point of view brings into sharp relief many of the issues that arise in employing transfer techniques when building electronic devices incorporating graphene. Understanding the transfer process on a conceptual physical level also provides insight into how to overcome these issues and design better graphene device fabrication processes.

Existing reviews tend to divide graphene transfer methods into etching and non-etching, based on whether graphene is separated from its substrate physically or whether the substrate is chemically etched away to create freestanding or polymer supported graphene. We have chosen not to divide the methods this way, but have instead considered the transfer process as consisting of two main parts: (1) delamination of graphene from its original substrate, and (2) relamination of graphene onto its target substrate (Figure 1). We have organized the review around these two steps since it makes apparent the fact that different steps in the transfer process are prone to induce different types of defects and impurities in the transferred graphene. This division also highlights the fact that an overall mitigation strategy should encompass all aspects of the process. Throughout the rest of this review, we will use the terms “original substrate” and “target substrate” to refer to the substrates from which and to which the graphene is transferred, respectively.

The remainder of this paper is structured as follows. In Section 2, we introduce the main challenges that arise during the graphene transfer process. We group them mainly by whether these issues are most likely to arise during delamination of graphene from its original substrate or during relamination onto its target substrate. In Section 3, we introduce a simple physical model for the adhesion of graphene to its substrate. In Section 4, we perform a survey of transfer strategies specifically for the graphene delamination step, focusing in subsections on each of the relevant variables from the model introduced in Section 3. In Section 5, we perform another survey of transfer strategies, this time focusing on the graphene relamination step. In Section 6, we conclude with a few words about possible future directions of research.

A note about terminology: in general, graphene does not explicitly form covalent bonds with its underlying substrates. Several covalently bound carbon morphologies are present during epitaxial growth of graphene on silicon carbide, but even in this case, the carbon overlayer is generally not considered graphene until no covalent bonds exist between it and the underlying “buffer layer” [15]. We will follow this terminology here, and only consider graphene to be a hexagonal two-dimensional form of carbon which does not explicitly form covalent bonds with the underlying substrate material. However, despite the absence of covalent bonding, the adhesion of graphene to its substrate can be quite strong, especially once close conformal contact has been achieved.

## 2. Issues Arising in Graphene Transfer

The promise of incorporating graphene into electronics runs up against several practical issues, but the two most important points related to graphene transfer are those of cleanliness and structural integrity. The superlative electronic properties of graphene are extraordinarily sensitive to chemical doping as well as crystal defects [16,17,18,19,20,21]. Ideally, the graphene transfer process would allow the placement of pristine graphene onto an arbitrary substrate with control over doping, wrinkling, tearing, or folding. However, extensive work has shown that existing transfer techniques can fall short of this ideal, with transferred graphene bearing significant residual metallic [22] contamination from polymer support [23,24,25] and extensive physical damage to the graphene (Figure 2).

The two main steps of graphene transfer are delamination from the original substrate and relamination onto the new target substrate. As mentioned above, the primary issues which arise during delamination are not necessarily the same as those which arise during relamination. In the case of CVD grown graphene, the original substrate is the metallic substrate on which the graphene was grown. Removal of this substrate using chemical etchants exposes graphene to metal ions, which can potentially dope the graphene, changing its electronic properties [29]. One study measured a residual metallic contamination of greater than 10^13^ atoms·cm^−2^, regardless of the extensiveness of the post-etching cleaning procedure [22]. Another study measured the effect of these residual metal ions leftover from transfer on graphene’s electrochemical properties, showing that these metallic species could have an electrocatalytic effect on certain reduction reactions [30]. The implication is that researchers must take care when reporting unexpected properties of graphene, as the actual explanation for those observations might lie with impurities introduced in the transfer and subsequent processing of graphene (rather than with the graphene itself).

In addition to chemical impurities introduced during delamination, physical damage of the graphene sheet might also occur. This damage can appear regardless of whether or not chemical etching is used as a delamination strategy. Mechanically, graphene is one of the strongest materials ever observed, with a single sheet having an elastic modulus of roughly 1 TPa, as observed by atomic force nanoindentation experiments [31]. However, it is important to remember that, regardless of superlatives, graphene is only 1 atom thick, and the forces in the aforementioned measurements which resulted in mechanical failure of the material were on the order of micronewtons. Realistically, this force can easily be exceeded in transfer procedures involving mechanical peeling from surfaces. Graphene floating freely on water without a support can also undergo mechanical damage [24]. Even with a polymer support, cracks and tears can be introduced [27,32]. Moreover, the surface tension of water (72 μN·mm^−1^) is high enough that formation and popping of bubbles, whether generated during electrochemical bubble transfer or during chemical etching of the substrate, can also exceed the mechanical failure threshold of graphene [33,34].

Relamination of graphene onto the target substrate presents its own set of unique issues to overcome. Foremost among these issues is contamination of the graphene arising from removal of an assistive polymer support. A number of reports have detailed the polymer residue left behind on graphene as well as different strategies for their removal. This problem is not unique to CVD-grown graphene; mechanically exfoliated graphene (Scotch Tape method) also often exhibits glue residue [35]. The most common polymer removal methods use some combination of solvent cleaning and thermal annealing under a reducing atmosphere [25,36]. Thermal annealing has the secondary benefit of eliminating volatile molecular species that might be trapped between the graphene and the target substrate after relamination. However, Kumar et al. found that thermally annealing graphene transferred using poly(methyl methacrylate) (PMMA) as a polymer support in a reducing atmosphere is found to increase strain and chemical doping of the graphene [24]. Lin et al. attribute this effect to the thermal breakdown of PMMA and possible covalent binding of breakdown product fragments to the graphene [23]. Vacuum annealing was shown to reduce, but not eliminate completely, these polymer residues [24,28,37].

Mechanical damage to graphene is also an issue during the relamination process, but one important challenge that is unique to relamination is control over wrinkling of graphene on the target substrate. As a thin sheet, graphene buckles and folds over itself easily, and it is extremely difficult to remove folds and wrinkles from graphene once it is deposited onto its final target substrate. Graphene wrinkles have unique electronic properties that may be exploited in some circumstances [38,39], but often it is desirable to eliminate them. While methods exist for eliminating wrinkles by judicious choice of target substrate [40], the most straightforward way to avoid wrinkle formation is never to introduce them in the first place. Doing so requires understanding the factors which cause wrinkles to appear during graphene relamination.

## 3. Physics of the Graphene-Substrate Interaction

### 3.1. General Overview

The graphene-substrate adhesion is mediated solely by van der Waals (vdW) forces. Dispersion forces, which are the weakest of the intermolecular forces, dominate this interaction in general [41], but depending on the substrate, the adhesion energy can also have contributions from ionic and covalent interactions [42]. However, due to graphene’s enormous surface area to volume ratio and the close conformation of a graphene sheet to its substrate, the additive nature of the forces renders them quite strong [43]. Whereas the vdW-force-mediated interaction energy per unit area between the planar surfaces of two extended bodies is proportional to 1/r^2^, where r is the distance between the surfaces, the areal interaction energy between an extended body and an infinitely thin plane of atoms is proportional to 1/r^3^ [44]. This difference in the exponent is important: at small r, it leads to much stronger adhesion between an ultrathin film and a substrate than in the case of a thick film. However, at larger r, it leads to weaker adhesion.

Moreover, the van der Waals interaction itself is intrinsically strong for graphene. The interaction strength is proportional to the integral of the imaginary part of the dielectric function, which is dominated by the DC conductivity of the material [45]. The high conductivity of graphene, a zero-bandgap semiconductor, thus translates to a higher interfacial interaction energy between graphene and its substrate than can be found in other 2D materials of lower conductivity. This factor impacts not only graphene’s contribution to the interaction energy, but the contribution of the substrate as well. One therefore expects that graphene will adhere more strongly to a metallic substrate than to an insulating substrate. A caveat here is that an insulating substrate containing a high density of local dipoles on the surface can induce local electrical polarization in the graphene, known in the literature as “charge puddles,” which may increase the adhesion of graphene to the substrate [46].

The physics of the graphene-substrate interaction informs a number of various strategies for removing graphene from one substrate and transferring it to another. Gathering the observations from the preceding paragraphs, we can write the interaction energy in a simple illustrative form:(1)$$E\propto -\frac{C_{s}C_{g}}{r^{3}}$$
where C_s_ and C_g_ represent interaction coefficients determined by the detailed atomic and electronic structure of the substrate and the graphene, respectively. Strategies for graphene transfer aim to weaken this interaction energy during delamination from the original substrate and strengthen it during relamination onto the target substrate, and numerous examples are present which target each of the variables C_s_, C_g_, and r (Figure 3). We will therefore subclassify transfer strategies in this review according to which variable they alter.

We stress that many, if not most, of the reports on graphene transfer methods in the literature employ multiple strategies throughout the process, so that individual papers often do not fall cleanly into a single category of strategies that we aim to discuss. Our association of a paper in this review with a certain strategy is not meant to imply that no other strategies are used in that paper; rather, it should be taken to mean that the paper in question is a particularly useful illustration of the strategy with which we have associated it.

### 3.2. Technical Details

The detailed results of the physics described in the previous subsection are given here. We follow Israelachvili in this section [44] and assume that the graphene-substrate interaction (1) is dominated by van der Waals interactions, (2) is additive, and (3) lacks retardation effects that become important at large graphene-substrate separations.

We begin by describing the origin of the 1/r^3^ dependence of the interaction energy in Equation (1). From the aforementioned assumptions, the potential energy between two pointlike molecules is:(2)$$w_{mol-mol}\left(r\right)=-\frac{C}{r^{6}}$$
where C is the specific dispersion constant for the interacting species. The assumed additivity of the interaction allows us to compute the potential energy between a molecule and a thick flat surface (which will ultimately represent our substrate) by integrating over an infinite half-volume to obtain the result:(3)$$W_{mol-sub}\left(D\right)=-\frac{\pi {C\rho }_{s}}{6D^{3}}$$
where ρ_s_ is the density of the substrate molecules and D is the distance between the molecule and the surface of the substrate. To obtain the graphene-substrate interaction, we first note that the interaction energy of two infinitely extended parallel surfaces will be infinite, so that we must look instead at per-unit-area interaction. Ordinarily, to obtain the interaction between two flat surfaces, we would assume their thicknesses t_1_ and t_2_ were much larger than the distance separating them: $t_{1}\approx t_{2}\gg D$. This would lead us to integrate a thin layer of the second surface over its total thickness in the z direction to obtain the familiar result:(4)$$W_{sub-sub}\left(D\right)=-\frac{\pi {C\rho }_{1}\rho_{2}}{6}\int_{z=D}^{z=\infty }\frac{\mathrm{dz}}{z^{3}}=-\frac{\pi {C\rho }_{1}\rho_{2}}{12D^{2}}$$
where ρ_1_ and ρ_2_ are the atom densities of substrates 1 and 2. However, in the case of a monolayer of graphene, integration in the z direction is not appropriate, and the graphene thickness t_g_ ≈ dz simply becomes a parameter to give an interaction energy of:(5)$$W_{gs}\left(D\right)=-\frac{\pi {C\rho }_{s}\rho_{g}t_{g}}{6D^{3}}$$
where ρ_g_ is the density of the graphene. This is where the 1/r^3^ dependence arises in Equation (1).

Equation (5) is likely more familiar to physicists when written in terms of Hamaker constants $A=\pi^{2}{C\rho }_{1}\rho_{2}$ [47]. The Hamaker constant in this case is dependent on the material properties of both graphene and substrate as well as any dielectric material in the gap between the graphene and the substrate. Explicitly:(6)$$A=\frac{6\pi^{2}{\mathrm{kT}\rho }_{1}\rho_{2}}{{\left(4\pi \varepsilon_{0}\right)}^{2}}\overset{\infty }{\underset{n=0,1,2,\ldots }{\sum }}\frac{\alpha_{1}\left(i\nu_{n}\right)\alpha_{2}\left(i\nu_{n}\right)}{\varepsilon_{3}^{2}\left(i\nu_{n}\right)}$$
where k is Boltzmann’s constant, T is temperature, α_1_ and α_2_ are the polarizabilities of species 1 and 2, and ε_3_ is the dielectric permittivity of the material in the gap between the species all at imaginary frequencies. We also note that the *n* = 0 term in the summation is to be halved [48,49]. The expression in Equation (6) is the detailed version of our simplified substrate and graphene interaction coefficients in Equation (1) and shows explicitly how the interaction energy depends on the material properties of each of the components in the system. Strictly speaking, the contributions from the graphene and the substrate are not as cleanly separable as we have indicated in Equation (1). However, for heuristic purposes, the dependence of the interaction energies on polarizability is more or less directly proportional.

One important point to note about Hamaker constants is that they can be negative, generally when the dielectric properties of the material in the gap are intermediate between those of species 1 and 2 [50]. A negative Hamaker constant implies a repulsive van der Waals interaction. This presents yet another possible strategy for separating graphene and its substrate: introduce a material in the graphene-substrate gap with intermediate dielectric properties, thus effecting a repulsion between graphene and substrate. To the best of our knowledge, no one has yet attempted this strategy specifically. Given the large range of values reported for graphene’s dielectric properties [51,52], this strategy may not currently be practical.

## 4. Transfer Strategies: Delamination of Graphene

Delamination of graphene proceeds when the adhesion interaction between graphene and its original substrate is weaker than the interaction between graphene and some separating force, be it a secondary facilitating substrate such as a polymer support, a layer of water intercalating between graphene and its substrate, or a direct transfer to another substrate with a stronger adhesion to graphene. Thus, according to our model in Section 3, strategies for successfully delaminating graphene from its original substrate will involve either weakening the intrinsic substrate interaction coefficient C_s_, weakening the intrinsic graphene interaction coefficient C_g_, or increasing the distance r between graphene and the substrate.

As noted in Section 2, the delamination of graphene from its original substrate can introduce residual metallic impurities from the substrate as well as tears and cracks through mechanical deformation of the material. Many of the transfer techniques we will discuss were developed in order to overcome these issues. We will use our graphene/substrate adhesion model to classify the techniques mechanistically to obtain a better understanding of how altering the graphene-substrate interaction in various ways mitigates or exacerbates specific imperfections introduced during the transfer process.

### 4.1. Strategies Changing the Substrate Interaction Coefficient

Since the dielectric function is so intimately related to conductivity, one strategy for weakening C_s_ is to chemically alter the substrate to decrease its conductivity. In the case of a metallic substrate, this involves either oxidation of the metal, possibly to include chemically etching and removing the metal, or reduction of the passivating metal oxide layer to a metal, possibly capped with an intercalated neutral layer. Alternatively, one may provide a secondary transfer substrate, such as a polymer support, which interacts with graphene more strongly than the original substrate. Since the issues introduced when using a polymer support generally do not arise until relamination and removal of the support, we will treat polymer supports more extensively in Section 5.

Decreasing the substrate interaction coefficient can mitigate or exacerbate certain issues. When the methods used are chemically destructive to the substrate, metal dopant contaminants can be a major concern. Non-etching methods such as bubble-free electrochemical transfer can mitigate these issues to some extent, but still might feature a low level of chemical etching. However, the benefit of simplicity and low cost of these etching types of transfer often outweigh the drawbacks of metallic contamination, and these methods have become the most popular techniques for delaminating CVD-grown graphene from a metallic substrate.

#### 4.1.1. Chemical Etching of the Substrate

Since no covalent bonds are formed between graphene and its substrate, chemical methods allow the clean removal of the substrate via etching without damaging the graphene. By far, the most common situation calling for graphene transfer is from the metallic growth substrate to an insulating substrate. In this case, aqueous oxidizers are typically used to convert the substrate to water-soluble metal salts. The earliest, and still most common, method for transferring graphene from a metallic growth substrate involves adding a thin support layer, prototypically poly(methyl methacrylate) (PMMA), onto the top side of the graphene and subsequently oxidatively etching the metal from the underside of the graphene using either ammonium persulfate or iron (III) chloride (Figure 4a) [27,53,54,55]. A similar early method exists for transferring graphene from a silicon oxide donor substrate, involving etching of the SiO_x_ with NaOH in water (Figure 4b) [56]. The PMMA support prevents the delicate single layer graphene from folding back on itself or breaking up during the metal etching process. Also, since the PMMA/graphene layer floats, the graphene can easily be retrieved from the water bath onto an arbitrary substrate. The polymer support can then be dissolved using acetone [54] or acetic acid [57], thus completing the graphene transfer.

#### 4.1.2. Non-Etching Methods to Weaken the Substrate Interaction Coefficient

Cabrero-Vilatela et al. used a combined approach of oxidizing the copper growth substrate in water and mechanically peeling graphene from the substrate with an atomic layer deposited (ALD) layer of Al_2_O_3_ and an adhesive (Figure 5a). The authors claim that use of the ceramic layer prevents contamination from a polymer support [58]. We point out, however, that other researchers report that ALD is only effective on graphene with a certain defect density; ALD on pristine graphene is ineffective due to the lack of seed sites to initiate atomic layer growth [59]. Grebel et al. employ a similar method using either alumina or hafnia deposited via ALD, but they also employ copper etching methods in their work [60].

Bubble-free electrochemical methods for transfer have also been put forward, with the reasoning that the H_2_ bubbles formed during bubble transfer (vide infra Section 4.3.2) can mechanically damage the thin, fragile graphene. Cherian et al. studied low-potential electrochemical methods where they described the mechanism of delamination as the reduction to metallic copper of the copper oxide passivating layer between copper and graphene, weakening the substrate [62]. Wang et al. extended this method by carbonating the electrolyte bath, coupling carbonate-based chemical reduction of copper oxide with its electrochemical reduction, with the ancillary goal of being able to recycle the copper growth substrate (Figure 5b–d) [61].

### 4.2. Strategies Changing the Graphene Interaction Coefficient

Just as in the case for C_s_, the most straightforward way to weaken C_g_ is to chemically modify graphene to decrease its conductivity. Whitener et al. showed that chemical hydrogenation of graphene decreases the conductivity of the material by several orders of magnitude, while simultaneously weakening the adhesion between graphene and substrates as diverse as metal growth substrates, silicon oxides, and polymers (Figure 6) [63]. Additionally, hydrogenation of graphene is reversible chemically and thermally [64,65,66,67,68], so that once the hydrogenated graphene is on the target substrate, it can be restored to pristine graphene, thereby completely avoiding the need for etchants or polymer support.

Physically, these techniques share a commonality with water-based delamination of a wide variety of nanomaterials [69,70,71,72,73]. The basic concept is that the separation of a 2D material and its substrate, which is generally energetically unfavorable, can be coupled with the interaction of the 2D material and the substrate with a third component (in this case, water) which is very energetically favorable. Hence, for example, when graphene is hydrogenated, the interaction with its substrate is weakened. Then, when the system is dipped in water, the water interacts strongly with both the substrate and the graphene to act as a wedge, easily and cleanly separating the graphene and the substrate. In addition, the hydrophobicity of the hydrogenated graphene [74,75,76,77] likely stabilizes the graphene on the water surface without the need for a polymer support. The hydrogen functionality acts akin to a surfactant, mimicking a technique that has been previously employed in aqueous exfoliation of bulk graphite [78].

As mentioned, the strategy of changing the graphene interaction coefficient can avoid the use of metal etchants as well as polymer supports, eliminating contamination from these two sources. In addition, the surfactant-like nature of hydrogen functionalization gives the graphene a tendency not to self-adhere. This might ultimately lead to less wrinkling. However, the lack of a support means that graphene is much more prone to mechanical damage during the transfer process. The removal of hydrogen functionality after relamination also represents an extra step in the graphene preparation process, which also ultimately decreases its attractiveness. The main advantage of this transfer strategy over others is the ability to functionalize graphene and have it maintain its functionality throughout the transfer process.

### 4.3. Strategies Increasing the Graphene-Substrate Distance

The interaction energy between graphene and its substrate falls off rapidly with distance. Thus, physically increasing the distance between graphene and its substrate is an effective strategy for non-destructive transfer of graphene. A number of prominent approaches in this category are represented in the literature. These include intercalation methods and electrochemical methods (sometimes referred to as “bubble transfers”) as well as mechanical and mechanochemical methods involving pressure-sensitive and thermal release adhesives.

Since these strategies do not strictly employ chemical etching of the original substrate, metallic contamination tends to be less of an issue for them. However, as mentioned above, electrochemical strategies might release a small amount of metal ions, which must be accounted for in high precision experiments. Regardless, the graphene transferred using these methods tends to be cleaner and more free from dopants than in the case where the substrate is etched away.

The main drawback of methods increasing graphene-substrate distance is that physical force is applied to the graphene to separate it from its original substrate. This force can cause tears and cracks to form in the graphene (e.g., Figure 2a). The main challenge for these strategies is therefore mitigating the separating force, which can be achieved in a number of ways. We discuss some of these innovative ways in more detail below, including directional etching of substrates [79], ensuring close conformal contact between adhesives and graphene [80], and using easy-to-remove original growth substrates such as liquid metals [81].

#### 4.3.1. Intercalation Methods

Intercalation methods seek to insert atoms and molecules in the space between the graphene and the substrate. Graphite intercalation compounds are well-known, especially with respect to alkali metal intercalation compounds [82]. Several intercalation methods have been demonstrated. Verguts et al. examined a bubble-free electrochemical delamination method and showed that water intercalation between the graphene and the substrate was crucial for effective separation of the layers. They further showed that intercalation inhibits relamination onto a target substrate [83]. This is to be expected, as intercalation increases graphene-substrate distance, facilitating delamination. However, relamination is favored when the distance between graphene and the target substrate is decreased, so deintercalation should be the goal for the relamination step.

Other more involved intercalation schemes have been explored. Recently, Guo et al. have demonstrated intercalative growth of silicon oxide between graphene and a ruthenium metal growth layer. The presence of silicon oxide increases the distance between the graphene and the metal substrate. However, the researchers did not use the separator to facilitate transfer; they used the intercalated silicon oxide as a new substrate on which to build graphene devices directly [84]. Ma et al. intercalated carbon monoxide gas between graphene and a platinum original substrate. Combined with a polydimethylsiloxane (PDMS) stamping and peeling method, they were able to cleanly remove graphene from the platinum in order to recycle the precious metal for further CVD graphene growth [85]. Ohtomo et al. were able to intercalate a thiol-based self-assembled monolayer (SAM) between graphene and its metallic original substrate [86]. The effect on decreasing adhesion energy was twofold here. First, the long hydrocarbon chain of the SAM provided a spacer over 1 nm in length to increase the distance between graphene and its substrate. Second, the interaction between the hydrocarbon SAM surface and the graphene was inherently weaker than the interaction between the metallic substrate and the graphene.

#### 4.3.2. Electrochemical Methods, including “Bubble Transfer”

Electrochemical methods pioneered by Wang et al. involve immersing the graphene on copper in water and applying a bias to the copper substrate (Figure 7a) [33]. This leads to water electrolysis and H_2_ bubble formation between the graphene and the metal, physically increasing the distance between the graphene and its substrate. Strictly speaking, as there is no water between the graphene and the metal to begin with, water must somehow intercalate between graphene and metal at the graphene edge [87], so these methods are also intercalation methods. In addition, since there is often some electrochemistry going on at the graphene-metal interface, some of these methods might also feature slight chemical etching. However, Gao et al. observed no metal ion contamination of bubble transferred graphene from platinum, pointing to the chief advantage of non-etching transfer methods: avoiding the introduction of metal ion impurities onto the graphene [87].

Bubble transfer can also be combined with other methods such as mechanical stamping (vide infra, Section 4.3.3). Chandrashekar et al. performed an electrochemical delamination coupled with a stamping method of graphene transfer (Figure 7b) [88]. The transfer of graphene from copper onto a flexible and transparent polymer (FTP) was carried out with the aid of an ethylene-vinyl acetate (EVA) coat. Delamination occurs via electrochemical hydrogen bubbling with the copper substrate acting as a cathode. Then, the graphene/EVA/FTP structure separates from the copper as H_2_ bubbles are generated between the graphene and copper substrate.

One interesting strategy to mitigate cracking is directional etching of the metal growth substrate. Zhang et al. performed copper etching in ammonium persulfate solution while applying a bias across two electrodes. The copper was observed to dissolve preferentially from the cathode-oriented side to the anode-oriented side. Presumably, this method mitigates cracking since the stress differential applied to the graphene is more uniform than when the copper is etched in random locations across the graphene when no bias is applied [79].

#### 4.3.3. Mechanical Methods

Mechanical methods rely on the van der Waals force between graphene and its initial substrate being weaker than the force between graphene and either its receiving substrate or an intermediate substrate that acts as a transfer aid. These intermediate substrates have generally consisted of adhesive polymers added to the top side of the graphene, to physically pull the graphene from the substrate. We saw in the section on etching strategies that several groups have examined the use of adhesive tapes in conjunction with etchant baths which explicitly remove the substrate. However, adhesives have also been used without etchant to transfer graphene from metallic or silicon carbide growth substrates. These adhesive techniques are quite varied. Popular techniques include the use of PDMS elastomer stamps [89,90] to peel graphene off its growth substrate and release it onto arbitrary substrates. An interesting example of mechanical-based transfer was demonstrated by Kim et al., to cleanly isolate single-layer graphene from a silicon carbide growth substrate. Epitaxial graphene grown from SiC tends to produce bilayer graphene “stripes” at step edges. To eliminate these stripes upon transfer, the team first deposited onto the graphene a strained nickel film produced by alternately vapor depositing and sputtering nickel onto the target. The strain and adhesion allowed for facile delamination of the graphene from the SiC. Then, to eliminate the bilayer graphene stripes, a second film of gold was added to the other side of the graphene. The gold-graphene adhesion energy was intermediate between the nickel-graphene and the graphene-graphene adhesion energy, allowing the gold to remove the stripes while leaving the continuous single layer graphene intact (Figure 8a) [8].

Stronger interactions than simple dispersion forces have also been exploited to separate graphene from its substrate. Lock et al. employed a diazonium-based adhesive to directly bind graphene sheets covalently, increasing the force that could be applied for separation of graphene from its substrate [91,92]. Seo et al. used an amine-rich viscoelastic polyethyleneimine (PEI) polymer gel as a transfer layer to facilitate mechanical delamination. The mechanism here is two-fold: first, the soft gel perfectly conforms to and coats the graphene, such that the PEI makes a high-surface-area contact with the graphene; and second, the PEI *n*-dopes the graphene such that the electrostatic adhesive interaction between the graphene and the PEI-GA layer exceeds that of the graphene-copper bond (38 J m^−2^). These two effects contribute to the complete wrinkle-free delamination of the graphene from its substrate (Figure 8b) [80]. The concept employed in all of these transfer techniques is that the interaction energy between the peeling agent and the graphene should be larger than the interaction energy between the graphene and the initial substrate.

Mechanical methods which do not use intermediate adhesive substrates are also known. In this case, the relevant interactions are the comparative magnitudes of the van der Waals force between graphene and its initial and final substrates. Fechine et al. used hot pressing with a variety of polymers to transfer CVD graphene directly from copper to the polymer [93]. They found that the increased contact area achieved by applying molten polymer to the graphene aided in increasing the strength of interaction between graphene and the polymer relative to the substrate. Another example of direct mechanical transfer involves using molten gallium as a liquid metal growth substrate for CVD graphene. Fujita et al. demonstrated low temperature (50–150 °C) growth of graphene at the interface between molten gallium and either sapphire or polycarbonate. The gallium was removed in the liquid state using a stream of N_2_ gas [81].

#### 4.3.4. Laser-Assisted Methods

Finally, a number of groups have recently demonstrated a novel way of transferring graphene across an air gap by focusing a laser pulse onto a graphene-coated substrate (Figure 9) [94,95,96,97,98]. A disk of graphene at the highest-fluence area on the substrate detaches and is pushed onto a new substrate. Depending on the nature of the experimental setup, the exact mechanism for delamination can vary. In general, though, it either involves trapped gases that expand to force the graphene away from the substrate [97], or potentially a photothermally-induced graphene-substrate lattice mismatch which weakens the interaction strength between the two species [94]. Spatial localization of the graphene transfer can be very precise using laser-assisted methods. However, the transfer often consists of many steps (Figure 9a), and damage can be introduced at several points during the transfer. Somewhat counterintuitively, however, Praeger et al. report that their laser-induced backward transfer experiment shows little sign of oxidative damage to graphene when performed at ambient pressure. This observation suggests that heat transfer from the graphene is sufficiently rapid to prevent the material from burning in the air under the high laser fluence [94].

### 4.4. Other Methods and Combinations

Incorporating a number of these ideas, Chandrashekar et al. devised an impressive method for roll-to-roll transfer of graphene from copper to a polymer by combining hot pressing of the polymer, resulting in favorable graphene-polymer interaction, with hot water delamination of the graphene from the copper [99]. Shivayogimath et al. employed the same idea using a simple and inexpensive office laminator for the hot-pressing and roll-to-roll transfer. This process consists of oxidatively decoupling the graphene from the catalytic surface, then laminating polyvinyl alcohol (PVA) on graphene’s surface. The strong adhesion between graphene and PVA layer supports the mechanical delamination of graphene from the copper substrate and largely damage-free, as indicated by mobility and Raman spectral data [100].

## 5. Transfer Strategies: Relamination of Graphene

Relamination of graphene as a separate step in the transfer process has seen far less specialized attention than delamination. However, the relamination process is extremely important to the quality of the final product, raising issues that are not typically seen during the delamination step of transfer. Wrinkling and folding, for instance, are far more likely to happen upon relamination of graphene than delamination. The presence of wrinkles can also lead to other defects: trapping residual polymer or metallic contaminants by shielding them from cleaning procedures, and causing tears and cracks by introducing strain into the graphene lattice [39].

Our graphene substrate interaction model suggests that the main factors affecting graphene adhesion are tuning of the intrinsic substrate interaction (namely, ensuring that the target substrate provides a stronger interaction than either the original substrate or a temporary polymer support), tuning of the intrinsic graphene interaction, and manipulation of the distance between graphene and substrate, where a close conformal contact between graphene and the target substrate will tend to mitigate issues arising specifically during relamination such as wrinkles.

### 5.1. Strategies Changing the Substrate Interaction Coefficient

Physically, relamination of graphene is simply the inverse of delamination, so that promoting strong adhesion to the target substrate is largely a matter of inverting the conditions that one would promote for weakening adhesion during delamination. In reality, the processes are not complete inversions of one another. For instance, instead of strengthening the substrate interaction coefficient for relamination, it is much more common to delaminate graphene onto a temporary substrate which strongly adheres to it and subsequently find ways to weaken the adhesion to this temporary substrate in favor of the permanent target substrate. This strategy manifests itself most clearly in the form of a temporary polymer support applied during graphene delamination, which is subsequently removed upon relamination.

However, support removal can leave polymer residues which change the properties of the graphene, and therefore a number of techniques have been developed to limit the initial amount of residue left behind by polymer support during transfer, with the goal of obviating the need for extended cleaning procedures. Ren et al. reported a modification of the standard chemical etching with polymer support technique in 2012, where they omitted the PMMA step completely, simply allowing unsupported graphene to float freely on the etchant bath [101]. However, the unsupported graphene is so fragile that this method requires minimizing mechanical disturbances, even avoiding vibration of the etching bath by mechanical pumps or people walking nearby. Therefore, in general, the advantages of using a polymer support to prevent mechanical damage and breakup of the graphene usually far outweigh the drawbacks of removing polymer residue from the transferred graphene.

#### 5.1.1. PMMA Removal and Cleaning

As mentioned above, the most widely used graphene transfer techniques employ a temporary polymer substrate. This is both to support the graphene mechanically when its original substrate is removed and to facilitate the transfer by being, in effect, a substrate which exhibits tunable adhesion to graphene. The most prominent example is the use of PMMA as a polymer support. While the PMMA adheres strongly to graphene when it is in a solid state, dissolving it in acetone causes its adhesion to drop dramatically and makes it easily removable from the graphene surface.

It has been shown that PMMA and other support polymers leave behind physisorbed and chemisorbed species after being dissolved off graphene with acetone. Lin et al. showed that PMMA residues remain even after annealing at high temperatures in forming gas [23]. Therefore, a significant amount of effort has gone into developing effective cleaning procedures for graphene [102,103]. Some evidence exists that PMMA can be thoroughly cleaned from graphene with careful annealing in air [104]. However, this method risks burning the graphene or, at the very least, introducing oxidative impurities onto the graphene lattice. Thus, extreme care must be taken with the conditions of air annealing.

In the same spirit, Liang et al. developed a “modified RCA” cleaning procedure [36], by adapting a silicon wafer cleaning procedure first introduced by Werner Kern in 1965 while working for RCA [105]. Their method consists of alternating basic and acidic hydrogen peroxide washes, and care must be taken to avoid gas formation associated with the decomposition of H_2_O_2_, which would agitate and possibly damage the single layer graphene. Sun et al. determined using Raman spectroscopy that the main PMMA-based contaminant remaining after thermal annealing was non-covalently bonded methoxycarbonyl side chain material, and they used this finding to develop an electrolytic cleaning technique [106]. The graphene acts as a cathode in an electrolytic cell, and the H_2_ bubbles generated from electrolyzed water aid in liftoff of the polymer residues, in a mechanism similar to the bubble transfer technique for lifting off graphene. Park et al. later showed that the formation of bubbles was not strictly necessary, by performing a bubble-free electrochemical cleaning procedure in nonaqueous solution [107].

#### 5.1.2. Releasing Layers as Polymer Support

Another technique to limit polymer residue is the use of releasing layers, which are polymer support layers composed of material that only weakly or tunably adheres to graphene. It has been shown that polymers such as polystyrene or polyisobutylene have relatively weak interaction with graphene [108]. When they are used as a support layer, etching of metallic substrates proceeds as normal, but adhering the graphene to its new target substrate is possible, even when that substrate is hydrophobic. This is due to the fact that the adhesion energy between graphene and the target substrate is more favorable than adhesion between graphene and the releasing layer.

This releasing layer idea can be taken a step further by using thermal release tape (Figure 10) [109,110]. This material binds strongly to graphene at low temperatures, but the adhesion becomes significantly weaker at higher temperatures. Etching-assisted direct transfer of graphene by hot-pressing or hot embossing is also quite effective, and can produce large-area sheets of graphene continuously in a roll-to-roll process (Figure 10b,c) [110,111,112,113]. Hot pressing is advantageous when the target substrate is rigid, as is the case with silicon wafers or many other ceramic semiconductors, which cannot be processed in a roll-to-roll fashion. The above-cited processes are all wet-transfer, meaning that even though adhesion is achieved between graphene and a target substrate, the copper growth substrate is still etched away chemically.

#### 5.1.3. Other Methods Relying on the Substrate Interaction Coefficient

Calado et al. showed that wrinkling of graphene was minimized when the target substrate was hydrophobic. They hypothesized that water plays an important role in the formation of wrinkles, and that wrinkles provide a conduit through which water drains from between graphene and target substrate upon relamination [73]. A hydrophobic substrate preferentially excludes water and presumably eliminates wrinkles in this manner. Kim et al. provided a generalized method for obtaining wrinkle-free graphene using a similar concept, by employing a low surface tension, hydrophobic, organic liquid, heptane, as a facilitating layer and a pseudo-substrate between graphene and the target substrate. This hydrophobic layer promoted wrinkle-free graphene while simultaneously enabling attachment of the graphene to the desired target substrate, regardless of that substrate’s surface energy [114].

### 5.2. Strategies Changing the Graphene Interaction Coefficient

As mentioned earlier with respect to delamination, the ability to chemically modify graphene gives us control over C_g_. Whereas, in general, covalent functionalization of the graphene decreases C_g_ and therefore weakens the adhesion of graphene to its substrate, removing covalent functionalities from chemically modified graphene can strengthen C_g_ and allow previously delaminated graphene to adhere tightly to a new substrate. Whitener et al. have shown that hydrogenated graphene that has been delaminated onto a water surface can be dehydrogenated in situ by briefly exposing the material to bromine gas. The relaminated dehydrogenated graphene then adheres strongly to a target substrate, without further delamination upon introduction of water [67]. A non-covalent method to change the graphene interaction coefficient was pursued by Kafiah et al., who performed electrostatic charging of graphene to ensure a strong interaction between it and a polymer target substrate [115]. This transfer strategy seems to be relatively underexamined in the literature thus far.

### 5.3. Strategies Decreasing the Graphene-Substrate Distance

One of the earliest graphene transfer methods to eschew a polymer support was demonstrated by Regan et al. in 2010, who reported a direct transfer of graphene from a copper growth substrate to a TEM grid by placing a drop of isopropanol between the grid and the copper substrate [116]. The evaporation of the isopropanol through the holes in the grid pulled the graphene into close contact with the grid. Chemical etching of the copper freed the graphene which was then adhered only to the TEM grid, effecting a polymer-free transfer. However, the holes in the grid of the target substrate were necessary to allow evaporation of the isopropanol. Without them, this type of transfer would likely have been accompanied by buckling and wrinkling of the graphene as the hydrophilic isopropanol attempted to drain away from the graphene substrate interface, similar to the wrinkle formation mechanism described in Section 5.1.3.

Avoidance of wrinkling of the 2D sheets during relamination has driven much of the advancement in transfer strategies seeking to decrease the graphene-substrate distance during relamination. In particular, many strategies focus on the use of very soft polymer supports which can be easily melted and are thus conformable simultaneously to flat graphene and to the target substrate [117,118,119]. Of these materials, paraffin in particular has the interesting feature of a comparatively low boiling point, allowing the removal of the support material by heating under low pressure (Figure 11a) [120]. A few researchers have taken this “soft polymer” approach to its logical conclusion. Belyaeva et al. demonstrated graphene transfer at the interface of aqueous solution and cyclohexane [26], while Zhang et al. performed a very similar experiment using n-heptane as the “support” [121]. In both cases, an organic liquid immiscible with water was used that has a high vapor pressure at room temperature. The Zhang study also examined use of a low concentration of cellulose dissolved in water as an anti-wrinkle additive to the copper etchant bath. A definitive study on the use of anti-wrinkle agents in graphene transfer, however, has not been performed.

A similar soft polymer approach was advanced by Zhang et al., who replaced PMMA in the standard transfer with a commercially available 3M Nexcare liquid bandage (LB) [123]. The mechanical properties of LB, such as its low elastic modulus of 85 MPa and low average molecular weight, allow for a less contaminated graphene and an extremely flat surface. Analysis of Raman spectra, electron mobility, and hole mobility demonstrate how graphene’s properties have improved with LB support compared to a PMMA support layer.

In the same vein is the use of pressure sensitive adhesive as a polymer support. This technique ultimately has its roots in the Scotch Tape micromechanical cleavage method that Novoselov et al. used to isolate graphene in 2004 [124]. Kim et al. found that, by using polyethylene terephthalate (PET)-supported pressure sensitive adhesive, they could etch the metallic growth substrate from graphene and transfer the graphene onto a new substrate by applying light pressure (Figure 11b) [122]. Again, the important factor here is that the intimate contact between graphene and the target substrate produced an interaction energy that overcame the adhesive interaction between the graphene and the pressure sensitive adhesive.

### 5.4. Combinations and other Strategies

An unusual single-wafer transfer technique employed a remarkable method to avoid introduction of wrinkles to the graphene. Gao et al. reported growth and transfer of CVD graphene on the same wafer by first growing graphene on an ultrathin layer of copper on a specially treated SiO_2_/Si wafer, and then etching away the copper chemically. This etching forms voids in the copper which adhere the graphene to the underlying SiO_2_ via capillary forces. Such a process would naturally form wrinkles where the voids connect the SiO_2_ and the graphene, but the authors avoid wrinkle formation by modulating the surface tension of the water through addition of isopropyl alcohol. This technique allows an ultrasmooth, single-wafer transfer of CVD graphene [125].

## 6. Future Directions

The transfer of graphene is an indispensable step in graphene-based device fabrication. As we have seen, a great deal of progress has been made to mitigate many of the issues that arise during graphene transfer. Delamination and relamination each bring their own challenges, and understanding them and graphene-substrate interaction on a mechanistic level enables a rational, holistic approach to solving problems associated with each step of transfer.

Each transfer strategy comes with benefits and drawbacks, and the most effective transfers appear to be the ones which combine the strengths of several different strategies to mitigate many issues at once. At least part of the future of graphene transfer at a basic research level will likely be continuing to evaluate new combinations of strategies for cleanliness and integrity of the final product. From the standpoint of incorporating graphene into commercializable devices, however, there needs to be an examination of the tradeoff between complicating the transfer process by combining many strategies and reducing cost by simplifying the process. This tradeoff could see great progress in the use of automation specifically for transfer. Tight control of conditions during transfer would provide reproducibility, and an automated system could perform tasks such as mechanical peeling much more slowly and systematically, thereby introducing less strain into the graphene and hence less mechanical damage. Other techniques which are very promising commercially include clean delamination of graphene and recycling of substrates. Chemical etching is destructive and adds to cost, but a simple recycling scheme could make the transfer process and ultimately, graphene incorporation into devices, more economical.

In a less pragmatic, but more fundamental way, studies on graphene transfer can advance basic graphene science by probing graphene-materials interactions systematically. The physics outlined in Section 3 points to several fundamental questions that graphene adhesion studies could help address. As mentioned there, the dielectric properties of single and multilayer graphene are not particularly well-understood, and the range of values that has been obtained for them is rather wide. Examining graphene adhesion to a variety of substrates in different media could provide information about graphene-specific Hamaker constants and increase our understanding of the dielectric properties of graphene. Other interesting effects in graphene-substrate interaction have been observed; for instance, graphene and other 2D materials have been found to screen the dielectric properties of their substrates, effectively shortening the range of van der Waals forces for those surfaces [126,127,128]. Depending on the nature of the electronic structure of the 2D material, this screening can be complete or incomplete. This screening can also extend to hydrogen bonding [129].

In principle, the model that we have introduced is applicable not only to graphene transfer, but to transfer of any 2D material. Van der Waals heterostructures display an astonishing versatility of properties, including superconductivity magic-angle twisted bilayer graphene [130], long-lived excitonic states in mixed transition metal dichalcogenides [131], and ferroelectricity in twisted bilayer boron nitride [132]. Currently, most of these heterostructures are either grown or transferred via a stamping method. However, commercialization might require a more robust and scalable process, and the progress already made on graphene transfer will inform the development of transfer in these other materials.

## Figures and Tables

**Figure 1 nanomaterials-11-02837-f001:**
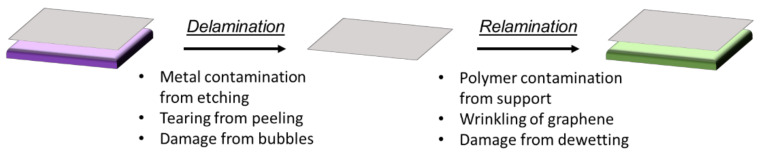
Pictorial representation of the two main stages of the graphene transfer process, delamination and relamination, accompanied by examples of issues that arise during each stage that can lead to degradation of graphene’s properties.

**Figure 2 nanomaterials-11-02837-f002:**
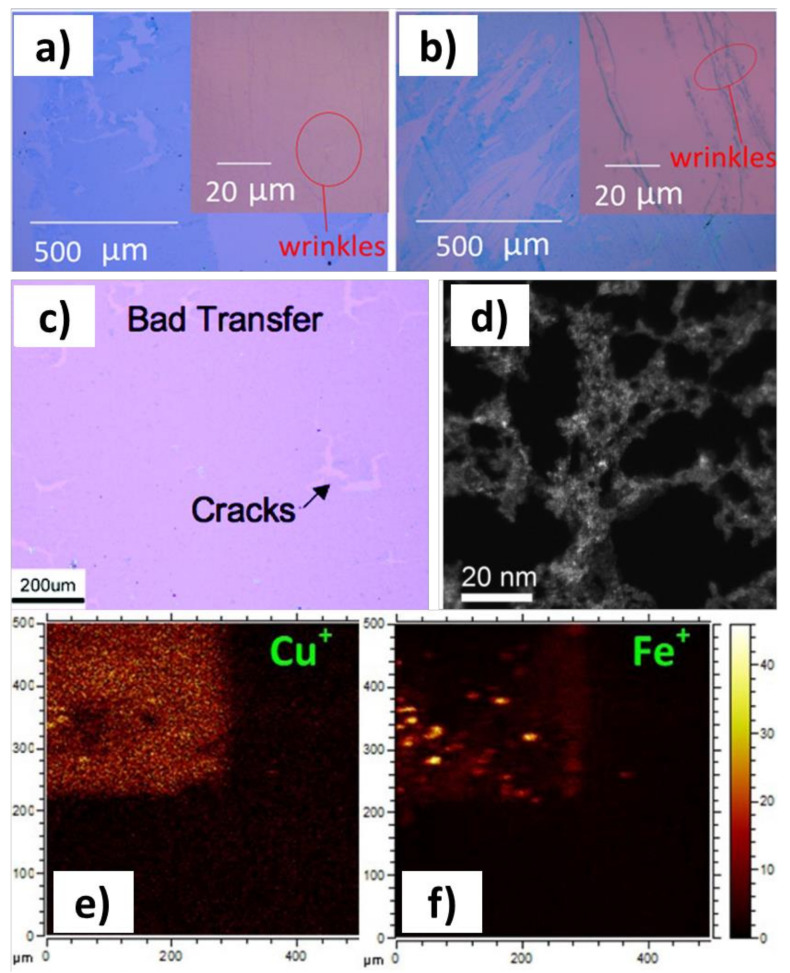
Defects and impurities introduced during graphene transfer. (**a**) Optical micrograph of tears introduced from stamping transfer of graphene. (**b**) Tears and wrinkling introduced during the relamination step liquid-liquid interface transfer of graphene. (**a**,**b**) Adapted with permission from Ref. [26]. Copyright 2016 American Chemical Society. (**c**) Cracking of graphene introduced during polymer supported etching-based graphene transfer. (**c**) Adapted with permission from Ref. [27]. Copyright 2009 American Chemical Society. (**d**) Transmission electron micrograph (TEM) image of residual polymer contamination on graphene from a polymer supported etching transfer. (**d**) Adapted with permission from Ref. [28]. Copyright 2017 Cambridge University Press. (**e**,**f**) Time-of-flight secondary ion mass spectrometry (ToF-SIMS) maps of copper (**e**) and iron (**f**) ions showing residual metallic contamination from etching-based transfer. (**e**,**f**) Adapted with permission from Ref. [22]. Copyright 2015 American Chemical Society.

**Figure 3 nanomaterials-11-02837-f003:**
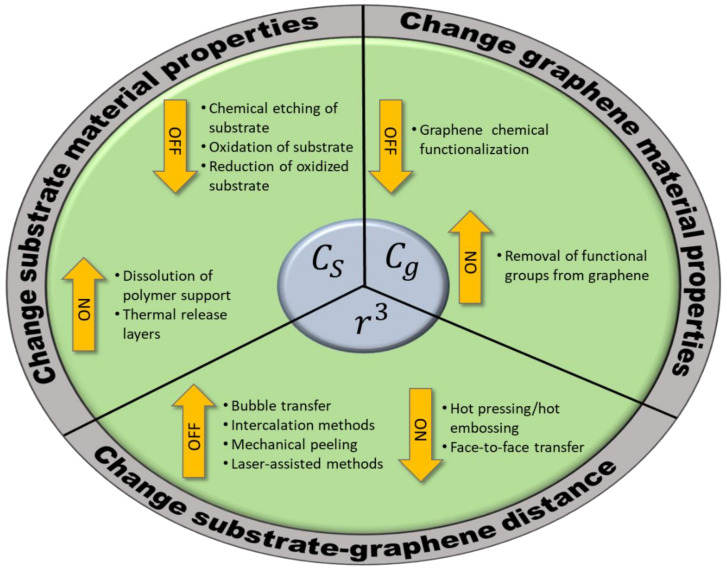
Graphical representation of different graphene transfer methods and the mechanisms by which they modulate the interaction between graphene and substrate. The central circle depicts the three variables which govern the graphene-substrate adhesion energy, and by changing each of the variables, one may promote either delamination (denoted “OFF”) or relamination (denoted “ON”) of the graphene on a substrate.

**Figure 4 nanomaterials-11-02837-f004:**
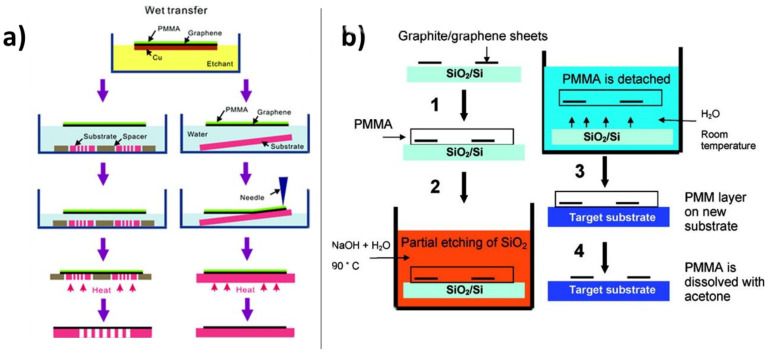
Chemical etching of underlying substrates to facilitate graphene transfer. (**a**) PMMA-assisted transfer of graphene from copper growth substrate. Adapted with permission from Ref. [53]. Copyright 2011 American Chemical Society. (**b**) PMMA-supported transfer with NaOH etching of underlying SiO_x_ substrate. Reprinted with permission from Ref. [56]. Copyright 2008 American Chemical Society.

**Figure 5 nanomaterials-11-02837-f005:**
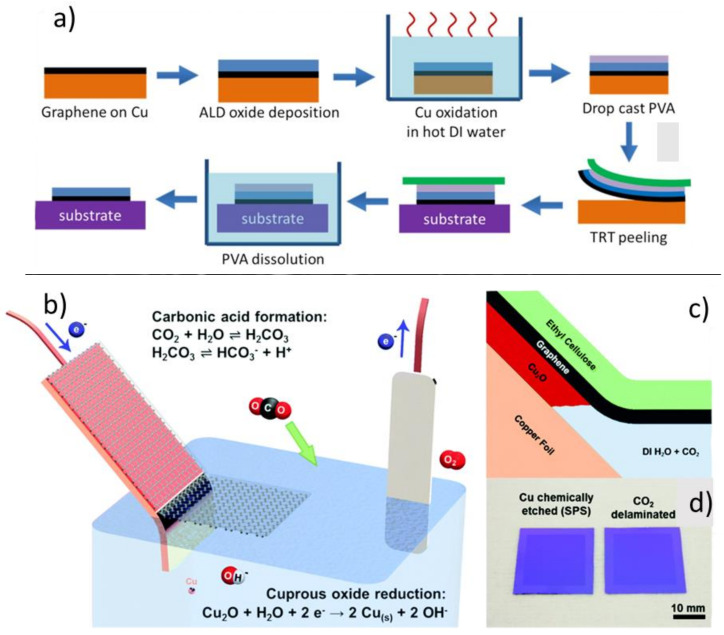
(**a**) Copper oxidation−assisted graphene delamination using ALD aluminum oxide to promote graphene peeling. Adapted under the terms of the Creative Commons Attribution 3.0 license from Ref. [58]. Copyright 2017 IOP Publishing Ltd. (**b**) Electrode setup for electrolytic removal of copper oxide passivating layer between graphene and the copper growth substrate. (**c**) Illustration of hypothesized chemical process for copper oxide dissolution. (**d**) Comparison of graphene transferred onto SiO_x_ using chemical etching of copper vs. non-etching substrate removal. (**b**–**d**) Adapted with permission from Ref. [61]. Copyright 2017 Royal Society of Chemistry.

**Figure 6 nanomaterials-11-02837-f006:**
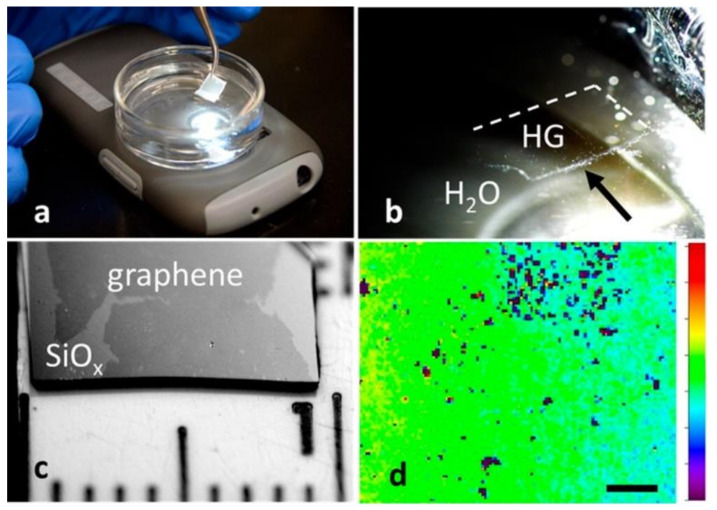
Hydrogen-facilitated graphene transfer. (**a**) Experimental setup of delamination of hydrogenated graphene directly into water. (**b**) Image of hydrogenated graphene floating on water. (**c**) Thermally restored graphene on a target SiO_x_ substrate. (**d**) Raman map of D peak of hydrogenated graphene showing retention of chemical functionality. Reprinted with permission from Ref. [63]. Copyright 2016 American Chemical Society.

**Figure 7 nanomaterials-11-02837-f007:**
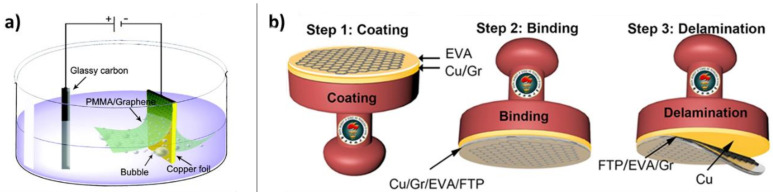
Bubble transfer methods. (**a**) Experimental electrode setup for electrochemical generation of hydrogen bubbles between the original substrate and the graphene to increase the graphene-substrate distance and weaken the adhesion energy. Adapted with permission from Ref. [33]. Copyright 2011 American Chemical Society. (**b**) An alternative experimental setup with non-PMMA support for graphene. Adapted under the terms of the Creative Commons Attribution 4.0 License from Ref. [88]. Copyright 2019, the authors.

**Figure 8 nanomaterials-11-02837-f008:**
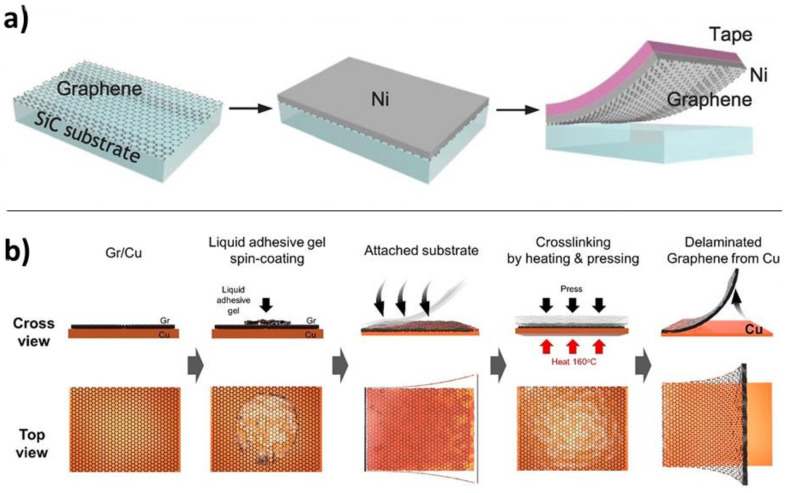
Examples of mechanical methods for facilitating delamination of graphene. (**a**) Nickel sputtering onto epitaxial graphene grown on SiC, followed by mechanical exfoliation of graphene. The nickel substrate adheres the graphene more strongly than the original SiC growth substrate. Adapted with permission from Ref. [8]. Copyright 2013 American Association for the Advancement of Science. (**b**) Liquid-phase adhesive-assisted delamination of graphene from an original copper growth substrate. The liquid gel is highly conformal to the graphene, augmenting its adhesive energy above that of the copper and allowing subsequent mechanical delamination. Adapted with permission from Ref. [80]. Copyright 2021 American Chemical Society.

**Figure 9 nanomaterials-11-02837-f009:**
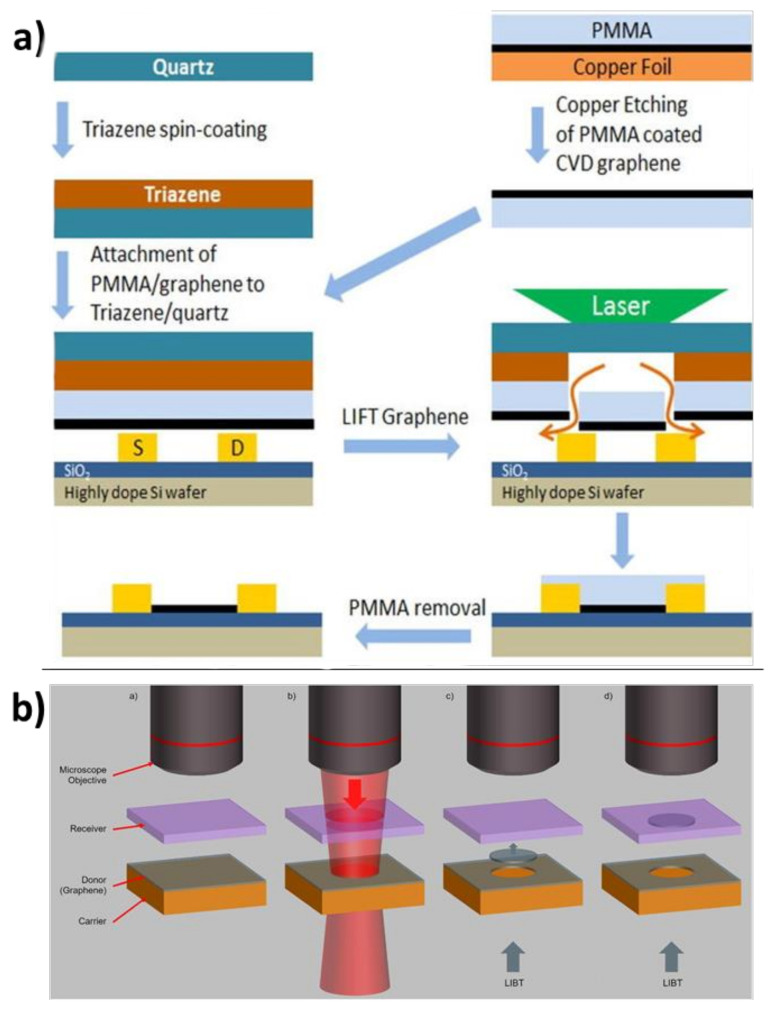
Examples of laser-assisted graphene transfer methods. (**a**) Laser-induced forward transfer experimental workflow. Reprinted with permission from Ref. [95]. Copyright 2017 American Institute of Physics. (**b**) Laser-induced backward transfer schematic, showing liftoff from the original substrate backward onto the target substrate. Reprinted under the terms of the Creative Commons Attribution 4.0 license from Ref. [94]. Copyright 2020 Elsevier.

**Figure 10 nanomaterials-11-02837-f010:**
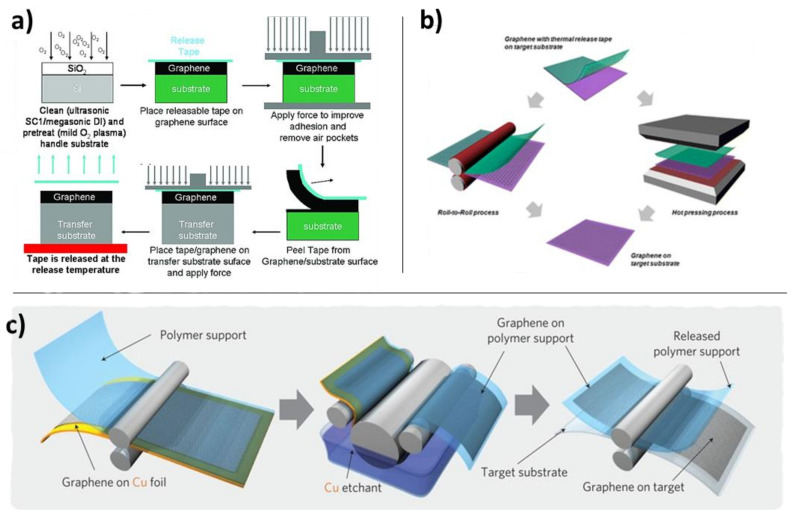
Thermal release tape strategies for temporarily weakening the graphene-substrate interaction to effect transfer to the target substrate. (**a**) Cold pressing with thermal release tape to remove air pockets. Reprinted with permission from Ref. [109]. Copyright 2010 American Chemical Society. (**b**) Roll-to-roll hot pressing with thermal release tape for large-area graphene transfer. Adapted with permission from Ref. [110]. Copyright 2012 American Chemical Society. (**c**) Large-area roll-to-roll graphene transfer with etching of original substrate and use of thermal release tape. Reprinted with permission from Ref. [112]. Copyright 2010 Springer Nature.

**Figure 11 nanomaterials-11-02837-f011:**
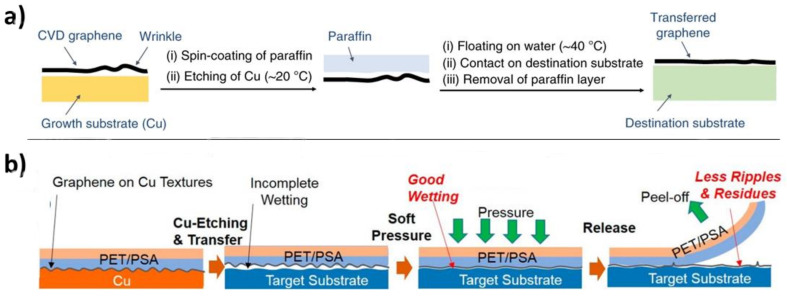
Using soft transfer substrates and close conformal contact with the target substrate to minimize wrinkling and ripples. (**a**) Paraffin-mediated graphene transfer. The low softening/melting point of paraffin enables conformal coating of the graphene, minimizing wrinkles. Adapted under the terms of the Creative Commons Attribution 4.0 license from Ref. [117]. Copyright 2019 Springer Nature. (**b**) Conformal contact between graphene and target substrate enabled by use of pressure sensitive adhesive and high wetting. Reprinted with permission from Ref. [122]. Copyright 2015 American Chemical Society.

## Data Availability

Not applicable.

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
