# Peer review of "Graphene Transfer: A Physical Perspective"

_nanomaterials, 2021, doi:10.3390/nano11112837_

Round 1
Reviewer 1 Report
In their paper, the Authors review the techniques for the transfer of graphene from the original substrate to the target substrate, examining separately the delamination and relamination phases and using a simple physical model for adhesion in order to classify the different methods.
The paper is enlightening, interesting, and very well written.
Therefore, I strongly suggest its acceptance, after some minor improvements.
In particular:
a) In the first lines of the Introduction, I suggest to add some further references to graphene and its application in electronic devices:
A. H. Castro Neto, F. Guinea, N. M. R. Peres, K. S. Novoselov, and A. K. Geim, "The electronic properties of graphene", Rev. Mod. Phys. 81, 109-162 (2009), DOI: 10.1103/RevModPhys.81.109;
F. Schwierz, "Graphene Transistors: Status, Prospects, and Problems", Proc. IEEE 101, 1567-1584 (2013), DOI: 10.1109/JPROC.2013.2257633;
P. Marconcini and M. Macucci, "Approximate calculation of the potential profile in a graphene-based device", IET Circuits, Devices Syst. 9, 30-38 (2015), DOI: 10.1049/iet-cds.2014.0003.
b) Regarding Figs. 4 and 5, not all the panels are referred in the text and the referred ones are cited in a different order with respect to that in which they are shown in the figures (moreover, at line 268, "5d" should appear instead of "5b"). I suggest to show the panels in the same order in which they are cited in the text and to cite all the panels (or the overall figures, if the Authors prefer);
c) On lines 352 (PDMS), 523 (RCA), and 623 (PET) the acronyms should be explicitly explained.
d) On line 510 the "as" before "its original" should be removed.
Author Response
Response to Reviewers
We thank all the reviewers for taking the time to read our manuscript and offer insightful comments and suggestions. Below are our responses to their reviews. Their comments are in black and our responses are in blue.
Reviewer #1
In their paper, the Authors review the techniques for the transfer of graphene from the original substrate to the target substrate, examining separately the delamination and relamination phases and using a simple physical model for adhesion in order to classify the different methods.
The paper is enlightening, interesting, and very well written.
Therefore, I strongly suggest its acceptance, after some minor improvements.
In particular:
- In the first lines of the Introduction, I suggest to add some further references to graphene and its application in electronic devices:
H. Castro Neto, F. Guinea, N. M. R. Peres, K. S. Novoselov, and A. K. Geim, "The electronic properties of graphene", Rev. Mod. Phys. 81, 109-162 (2009), DOI: 10.1103/RevModPhys.81.109;
F. Schwierz, "Graphene Transistors: Status, Prospects, and Problems", Proc. IEEE 101, 1567-1584 (2013), DOI: 10.1109/JPROC.2013.2257633;
P. Marconcini and M. Macucci, "Approximate calculation of the potential profile in a graphene-based device", IET Circuits, Devices Syst. 9, 30-38 (2015), DOI: 10.1049/iet-cds.2014.0003.
Citations have been added at appropriate places.
- Regarding Figs. 4 and 5, not all the panels are referred in the text and the referred ones are cited in a different order with respect to that in which they are shown in the figures (moreover, at line 268, "5d" should appear instead of "5b"). I suggest to show the panels in the same order in which they are cited in the text and to cite all the panels (or the overall figures, if the Authors prefer);
We have rearranged the figures to match the order in which they are referenced in the text.
- c) On lines 352 (PDMS), 523 (RCA), and 623 (PET) the acronyms should be explicitly explained.
Acronyms have been addressed, thanks!
- d) On line 510 the "as" before "its original" should be removed.
We have corrected the typo, thanks!
Reviewer 2 Report
It is an interesting review paper, authors analyzed the literature describing graphene transfer methods developed over the last decade. We present a simple physical model of the adhesion of graphene to its substrate, and we use this model to organize the various graphene transfer techniques by how they tackle the problem of modulating the adhesion energy between graphene and its substrate. Authors consider the challenges inherent in both delamination of graphene from its original substrate as well as relamination of graphene onto its target substrate, and simple model can rationalize various transfer strategies to mitigate these challenges and overcome the introduction of impurities and defects into the graphene. The analysis of graphene transfer strategies concludes with a suggestion of possible future directions for the field. It can be accepted after minor revision.
Recent review on graphene plasmon can be cited in the introduction to show the important of graphene transfer for the graphene devices, Reviews in Physics, 2021, 6, 100054..
Author Response
Response to Reviewers
We thank all the reviewers for taking the time to read our manuscript and offer insightful comments and suggestions. Below are our responses to their reviews. Their comments are in black and our responses are in blue.
Reviewer #2
It is an interesting review paper, authors analyzed the literature describing graphene transfer methods developed over the last decade. We present a simple physical model of the adhesion of graphene to its substrate, and we use this model to organize the various graphene transfer techniques by how they tackle the problem of modulating the adhesion energy between graphene and its substrate. Authors consider the challenges inherent in both delamination of graphene from its original substrate as well as relamination of graphene onto its target substrate, and simple model can rationalize various transfer strategies to mitigate these challenges and overcome the introduction of impurities and defects into the graphene. The analysis of graphene transfer strategies concludes with a suggestion of possible future directions for the field. It can be accepted after minor revision.
Recent review on graphene plasmon can be cited in the introduction to show the important of graphene transfer for the graphene devices, Reviews in Physics, 2021, 6, 100054..
Citation has been added, thanks!
Reviewer 3 Report
The manuscript is a very valuable review of the current state of knowledge and methods on the transfer of graphene between different substrates. I have only some questions regarding the simplified model proposed by the authors for the adhesion of graphene on the surface of the substrate. I find it very useful but:
(1) in Page 5 authors write the following: ”… the vdW-force-mediated interaction energy per unit area between the planar surfaces of two extended bodies is proportional to 1/r^2, where r is the distance between the surfaces, the areal interaction energy between an extended body and an infinitely thin plane of atoms is proportional to 1/r^3”.
I agree with this because for two flat plates the vdW interaction potential energy E can be approximated as
E(r)=-A/(12 \pi r^2), where A is representing a Hamaker constant. Then, why there is 1/r^3 in Eq.(1)?
(2) I am convinced that in this manuscript a short note on the vdW interactions would be helpful for the general readership. Notice, that generally the value of the Hamaker constant A is positive but in the case of the third medium with a particular value of the dielectric constant and index of refraction it can become negative or zero.
Author Response
Response to Reviewers
We thank all the reviewers for taking the time to read our manuscript and offer insightful comments and suggestions. Below are our responses to their reviews. Their comments are in black and our responses are in blue.
Reviewer #3
The manuscript is a very valuable review of the current state of knowledge and methods on the transfer of graphene between different substrates. I have only some questions regarding the simplified model proposed by the authors for the adhesion of graphene on the surface of the substrate. I find it very useful but:
(1) in Page 5 authors write the following: ”… the vdW-force-mediated interaction energy per unit area between the planar surfaces of two extended bodies is proportional to 1/r^2, where r is the distance between the surfaces, the areal interaction energy between an extended body and an infinitely thin plane of atoms is proportional to 1/r^3”.
I agree with this because for two flat plates the vdW interaction potential energy E can be approximated as
E(r)=-A/(12 \pi r^2), where A is representing a Hamaker constant. Then, why there is 1/r^3 in Eq.(1)?
We thank the reviewer for their comment. The inverse square distance dependence holds for the per-unit-area interaction between the flat surfaces of two infinite half-volumes. In our model, we consider instead the interaction of an infinitely thin planar sheet with an infinite half-volume. Normally, one would integrate twice over half-space, but because the graphene is planar, there is no second integral over the depth of the graphene (the graphene thickness can, however, be included as a parameter). This is the fundamental reason for the 1/r3 term in the model. We have included an elaboration on this point in the manuscript.
(2) I am convinced that in this manuscript a short note on the vdW interactions would be helpful for the general readership. Notice, that generally the value of the Hamaker constant A is positive but in the case of the third medium with a particular value of the dielectric constant and index of refraction it can become negative or zero.
This is a great idea. We have tried to strike a balance between an in-depth technical discussion of the physics of van der Waals interactions and an accessible introduction to a simple model for graphene-substrate adhesion for non-specialists. To that end, we have split section 3 on the physics of graphene-substrate interactions into two parts: a general overview for the non-specialist and a technical discussion for the specialist. In the technical discussion, we derive the inverse-cube result above as well as discussing the relationships between Hamaker constant, polarizability, and permittivity of the interfering medium to bring out the rationale behind our simple model more clearly for those interested.